# Occurrence and Estimated Daily Intake of Cortisone and Cortisol in Aquatic Food from China TDS

**DOI:** 10.3390/foods11213481

**Published:** 2022-11-02

**Authors:** Yi Yang, Jiachen Shi, Jie Yin, Yunjia Yang, Bing Shao, Jing Zhang

**Affiliations:** 1Beijing Key Laboratory of Diagnostic and Traceability Technologies for Food Poisoning, Beijing Center for Disease Control and Prevention, Beijing 100013, China; 2School of Public Health and Family Medicine, Capital Medical University, Beijing 100089, China

**Keywords:** cortisone, cortisol, aquatic products, estimated dietary intake

## Abstract

Glucocorticoids (GCs) widely exist in animal food including aquatic food. This study aimed to survey the occurrences of cortisone and cortisol in aquatic food and the estimated daily intake (EDI) of cortisone and cortisol due to different habits of aquatic food consumption. The mean levels of cortisone and cortisol in freshwater fish purchased from market were 14.59 μg/kg and 69.15 μg/kg, respectively, which were markedly higher than the levels in marine fish. A test using *Zebrafish* was performed to compare the concentration of GCs by different killing methods. The results suggested that physically traumatic killing methods are one of the reasons why the levels of GCs in freshwater fish were higher than those in marine fish. The concentrations of cortisone and cortisol in composite aquatic food samples from 12 provincial districts of the fourth China Total Diet Study (TDS) were 0.72~15.75 μg/kg and 4.90~66.13 μg/kg, respectively, which were positively correlated with the distance from the coastline. Further, the correlation coefficient between the levels of cortisone and cortisol in aquatic food and the percentages of freshwater fish consumption were 0.758 (*p* < 0.01) and 0.908 (*p* < 0.01), respectively. There was a significant positive correlation between the levels of cortisone and cortisol in aquatic food in the fourth TDS and the percentages of freshwater fish consumption. The calculated average EDIs of cortisone and cortisol from aquatic food in the fourth TDS were 0.16 μg/d and 0.72 μg/d, respectively.

## 1. Introduction

Over past decades, endocrine-disrupting compounds (EDCs) such as steroid hormones in food and the environment have attracted global concern because of their potential adverse effects [1,2,3,4]. Glucocorticoids (GCs) are an important class of endocrine-disrupting steroids. Natural and synthetic GCs act on the glucocorticoid signaling pathway and are involved in the regulation of energy metabolism, immune functions, stress responses and behavior control. Generally, GCs are used as anti-inflammatory and immunosuppressive therapies in clinical medicine [5]. However, high doses or long-term use of GCs can have side effects, such as osteoporosis, hypertension, obesity, acute myocardial infarction and infections [5,6,7]. Some research has indicated that GCs can regulate food-choice behavior in humans and further affect body weight [8]. Michael et al. suggested that GCs regulate the reproductive system by the hypothalamic–pituitary–gonadal axis, which was confirmed at the cellular level [9]. In addition, a new study suggested that higher concentrations of cortisone and cortisol in hair were inversely associated with memory and global cognition in older adults [10]. However, less attention has been focused on GCs in food and the environment than on estrogen and androgens.

Cortisone and cortisol are two important natural GCs biosynthesized in mammals, poultry and fish [11]. In order to estimate the potential risk to humans, it is important to investigate the concentration of naturally occurring hormones in food. In recent years, there have been increasing numbers of investigations on the concentrations of natural GCs in food. The average levels of cortisone and cortisol in Swiss Holstein cow milk were 112 and 235 ng/kg, respectively [12]. In our previous study, the average levels of cortisol in pork, beef, milk and shrimp were 15.65, 5.18, 0.43 and 1.40 μg/kg, respectively [13]. There were significant deviations in concentrations in different foods. The median concentrations of cortisone and cortisol across 37 fish samples were 9.8 μg/kg and 280 μg/kg, respectively [14]. It was speculated that the level of GCs in aquatic products would be higher than that in cattle or poultry. Thus, aquatic products would be one of the major dietary sources of GCs. However, few studies have focused on GCs in aquatic products.

In this study, the occurrence of cortisone and cortisol in aquatic food was investigated. Further, the estimated daily intake (EDI) of cortisone and cortisol due to different habits of aquatic food consumption was also investigated.

## 2. Materials and Methods

### 2.1. Chemicals and Reagents

HPLC-grade methanol and acetonitrile were purchased from Scharlau Chemic S.A. (Barcelona, Spain). Formic acid (99%) was obtained from Acros Organics (Morris Plains, NJ, USA). Deionized water was obtained from a Milli-Q system (Millipore, Bedford, MA, USA). Standards of cortisone and cortisol were purchased from Sigma (St. Louis, MO, USA). An internal standard, cortisol-d_3_, was obtained from Cambridge Isotope Laboratories (Andover, MA, USA). All standards were stored at −20 °C. Glucuronidase/arylsulfatase was purchased from Helix pomatia (Roche Diagnostics GmbH, Mannhein, Germany). GCB SPE cartridges (500 mg, 6 mL) and ENVI-carb cartridges (500 mg, 6 mL) were purchased from Supelco Co. (Bellefonte, PA, USA). NH_2_ SPE cartridges (500 mg, 6 mL) were purchased from Waters Co. (Milford, MA, USA).

Stock solutions (1 mg/mL) of cortisone and cortisol were prepared by dissolving 10 mg of target compound in 10 mL methanol, respectively, and these stock solutions were stored at −20 °C. The stock solution (1 mg/mL) of internal standard was prepared by dissolving 10 mg of cortisol-d_3_ in 10 mL methanol. A total of 100 μL stock solution of cortisone and 100 μL stock solution of cortisol were added into a 10 mL volumetric flask and diluted to fixed volume with methanol, and then the mixed working standard was obtained. The working solution of internal standard was diluted by stock solution of internal standard using methanol. Working solutions at serial concentrations were obtained by diluting aliquots of mixed working standard and working solution of internal standard with methanol. Each working solution contained 100 μg/L of cortisol-d_3_.

### 2.2. Zebrafish Culture

*Zebrafish* (*Danio rerio, AB strain*) were obtained from the Department of Biological Sciences and Biotechnology, Tsinghua University (Beijing, China). The fish were cultured in a flow-through system with charcoal-dechlorinated filtered tap water at 27 ± 1 °C with a light–dark cycle of 14:10 h. Adult fish were fed freshly hatched *Artemia nauplii* (Fengnian Aquaculture Co., Ltd., Tianjin, China) twice and flake food (Tetra, Melle, Germany) once daily.

### 2.3. Killing of Zebrafish

In this study, two killing methods were applied to *Zebrafish*. In the first method, 20 *Zebrafish* were put into an ice-bath for 10 min to kill them without physical trauma. Second, 20 *Zebrafish* were sacrificed by rapid decapitation. After being killed, the *Zebrafish* were homogenized, and the concentrations of cortisone and cortisol were measured.

### 2.4. Sample Collection

#### 2.4.1. Samples from Local Market

In this study, 26 marine fish (5 *Small yellow croaker*, 8 *Pomfret*, 6 *Ribbon* and 7 *Cod*) samples and 29 freshwater fish (6 *Carp*, 6 *Chub*, 8 *Grass carp* and 9 *Crucian*) samples were bought from a local supermarket in Beijing, China by the method of random sampling. The species and weight of each fish sample were recorded. The body weights of *Carp*, *Chub*, *Grass carp*, *Crucian*, *Small yellow croaker* and *Pomfret* were 1465–2130 g, 46–87 g, 2105–2455 g, 510–810 g, 328–553 g and 890–1560 g, respectively. The weights of *Ribbon* and *Cod* were not available since these fish were sold in segments. All fish samples were processed within 4 h of arriving at the laboratory. 

#### 2.4.2. Aquatic Foods Samples from TDS 

The consumption survey and sample collection for the fourth China Total Diet Study (TDS) were carried out in 12 provincial-level administrative divisions (PLADs), including Heilongjiang (HLJ), Liaoning (LN), Hebei (HeB), Henan (HN), Shanxi (SX), Ningxia (NX), Fujian (FJ), Hubei (HuB), Sichuan (SC) and Guangxi (GX) and Shanghai (SH). Food samples classified into 12 categories (e.g., cereals, aquatic foods, legumes, potatoes, meats, eggs) were collected from local markets, grocery stores and rural households, and were prepared and cooked according to the local consumption patterns in each site. The samples were then homogenized in a blender, with the proportion of each kind of food being weighted according to the average daily consumption of each PLAD [15,16]. Hence, 12 composite aquatic foods samples were obtained and stored at −20 °C until analysis. 

### 2.5. Sample Preparation

The method of sample preparation has been described in our previous paper in detail [13]. Briefly, five grams of the homogenized sample was transferred into a 50 mL polypropylene centrifuge tube, where 100 μL cortisol-d_3_ internal standard solution and 10 mL acetate buffer (0.2 mol/L, pH 5.2) were added. After homogenization, 100 μL glucuronidase/arylsulfatase was added, and the mixture was incubated overnight at 37 °C. Then, after the mixture had been allowed to cool to room temperature, 25 mL methanol was added and mixed with a vortex stirrer for 2 min. Then, the mixture was centrifuged at 10,000 rpm for 10 min at 4 °C, and the supernatant was decanted into a 250 mL beaker, diluted to 125 mL with ultrapure water. The solution was purified and enriched by GCB and NH_2_ SPE cartridges followed by eluting with dichloromethane/methanol (*v*/*v*, 70:30). The eluent was blown dry under a weak nitrogen stream, and the residue was dissolved in 1 mL methanol/water (*v*/*v*, 50:50) for LC-MS/MS analysis.

### 2.6. LC-MS/MS Measurement

The analysis method was referred to in our previous study [13]. Analysis was performed on a Waters Acquity UPLC^TM^ system coupled with a Quattro Premier XE mass spectrometer (Waters Corp., Milford, MA, USA). The LC column was an Acquity UPLC^TM^ BEH C18 column (100 mm × 2.1 mm, 1.7 μm) from Waters Co. (Milford, MA, USA). The column oven was set at 40 °C, and the injection volume was 10 μL. The mobile phases were A (water containing 0.1% formic acid) and B (methanol). Starting at 50%, the proportion of A was decreased to 36% from 0 to 8 min and then decreased to 16% from 8 to 11 min. Then, the proportion of A was decreased to 0% at 12 min and maintained at that level for the rest of the 15 min period. Finally, A was reset to 50% at 15.5 min. The flow rate was 0.3 mL/min.

## 3. Results and Discussion

### 3.1. Method Validation

In this study, the linearity, sensitivity, accuracy and precision were evaluated for the developed method. This method achieved good linearity over 0.03~100 μg/L. The correlation coefficients (*r*^2^) of cortisone and cortisol were higher than 0.999. The LODs of this method were 0.03 μg/kg and the LOQs were 0.1 μg/kg. The recoveries of cortisone and cortisol ranged from 89.6% to 93.2% with an RSD of 4.9% to 10.5%, which indicated that the method had good performance. 

### 3.2. Concentrations of Cortisone and Cortisol in Marine Fish and Freshwater Fish

Twenty-six marine fish and twenty-nine freshwater fish samples were purchased from a local supermarket. All freshwater fish were washed with water and weighed, and the marine fish were allowed to thaw at room temperature and were weighed before analysis. All fish samples were killed in the manner in which fish are usually slaughtered for general consumption. In brief, the freshwater fish were sacrificed by rapid decapitation, while the marine fish were used after thawing directly. Edible muscles were collected from these fish samples and then homogenized for further pretreatment as described in Section 2.5 and for LC-MS/MS analysis to measure the concentrations of cortisone and cortisol. As listed in Table 1, the mean levels of cortisone and cortisol in the marine fish were 0.76 μg/kg and 3.32 μg/kg, respectively. The levels of cortisone and cortisol in the freshwater fish were 14.59 μg/kg and 69.15 μg/kg, respectively. Thus, the levels of both cortisone and cortisol were markedly higher in the freshwater fish than in the marine fish. Chen detected 16 hormone residues in the aquatic products, and cortisol was not found in *Large yellow croaker*, a kind of marine fish, while the concentrations of cortisol in *Grass carp* and *Cyprinoid*, two kinds of freshwater fish, were 5.9–204.1 μg/kg and 12.3–105.6 μg/kg, respectively [17]. The concentrations of cortisol in seven kinds of freshwater fish were 7.7–45.3 μg/kg, as measured by the LC-Qtof-MS method [18], but cortisol was not detected in marine fish. The concentrations of GCs were similar to our study. Therefore, it was confirmed that the concentrations of cortisol and cortisone in freshwater fish were commonly higher than that in marine fish. 

### 3.3. Effect of Killing Methods on the Levels of Cortisone and Cortisol in Zebrafish

Some previous studies have suggested that the high concentrations of hormones in freshwater fish were due to the bioaccumulation of hormones in the environment [14]. However, GCs are a kind of stress hormone. The concentrations of GCs in humans or animals are related to stress effects. The high concentrations of GCs in the freshwater fish may be related to not only the bioaccumulation but also the stress before cooking, which could be due to the killing method. 

Generally, marine fish are frozen immediately after being caught, and are then delivered to consumers through a cold food chain. In contrast, freshwater fish are usually raised in tanks and sold to consumers at supermarkets or aquatic product markets and are usually killed by cutting the stomach or cutting off the head. Thus, before being cooked, marine fish are usually killed without physical trauma, whereas freshwater fish are usually killed by physical trauma. The secretion of cortisone and cortisol is related to the stress response in humans and animals. Hopkins et al. suggested that the level of cortisol in the blood of hellbenders was approximately 1.0 ng/mL immediately after capture and 4.5 ng/mL after handling stress [19]. It was postulated that the levels of cortisone and cortisol would be increased by physically traumatic killing methods. Furthermore, it was speculated that the physical trauma of slaughter was one of the reasons for the high concentrations of cortisone and cortisol in freshwater fish. In this study, the effect of different killing methods on the concentrations of cortisone and cortisol were investigated using *Zebrafish*, a commonly used fish model in toxicological and pharmacology studies [20,21].

First, 40 male and 40 female fish were randomly divided into four groups, including two groups of males and two groups of females. Then, one group of male fish and one group of female fish were sacrificed by ice-bath immersion. The other two groups were sacrificed by decapitation. After being sacrificed, all fish in each group were fully homogenized, and the concentrations of cortisone and cortisol were measured. As shown in Figure 1, the concentrations of cortisone and cortisol were significantly different in the different groups. In the male *Zebrafish* sacrificed by ice-bath immersion, the concentrations of cortisone and cortisol were 2.04 μg/kg and 12.81 μg/kg, respectively. The concentrations of cortisone and cortisol were 2.87 μg/kg and 31.09 μg/kg, respectively, in the male *Zebrafish* killed by decapitation. The concentrations of cortisone and cortisol in male *Zebrafish* killed by decapitation were 1.4 and 2.4 times higher than those in male *Zebrafish* killed by ice-bath immersion. Similarly, the concentrations of cortisone and cortisol in the female *Zebrafish* killed by decapitation were 2.5 and 2.3 times higher than those in the female *Zebrafish* killed by ice-bath immersion, respectively. The hypothalamic–pituitary–adrenal (HPA) axis is known to be activated when humans experience stress. Furthermore, the activation of this axis causes GCs to be secreted from the adrenal glands. The stress response in fish is similar to that in humans and is characterized by the activation of the hypothalamus–pituitary–interrenal (HPI) axis [22]. Stress also induced a significant increase in the level of cortisol in the blood of fish [23]. In this study, killing by decapitation induced a physical stress response. Therefore, the level of GC was immediately increased in the process of killing by decapitation because the HPI axis was activated by the stress response. These above results indicated that decapitation increased the levels of cortisone and cortisol. Furthermore, this result confirmed our hypothesis that a physically traumatic killing method is one of the reasons why the levels of cortisone and cortisol in freshwater fish are higher than those in marine fish.

### 3.4. Occurrence of Cortisone and Cortisol in Aquatic Foods in the TDS

TDS, as recommended by the WHO, is an important method for studying population dietary nutrition and food safety [24,25], and it has been carried out in more than 20 countries and regions. The fourth China TDS surveyed 1089 household (approximately 4320 persons) including different age groups from different regions. All survey points could represent the general dietary habits and actual dietary structure of residents of each province. In the TDS, all food samples were classified into 12 categories, in which aquatic food and aquatic food products were included. In each province, aquatic food samples were cooked and prepared according to the local dietary patterns and cooking methods [16]. Therefore, the survey data and samples of the TDS reflected the dietary situation of Chinese people in different regions over a long period of time. In contrast to the average or median level of GCs in fish, the concentration of GCs in the TDS sample for calculating the estimated daily intake was more reasonable. In this study, the EDI level of cortisone and cortisol from aquatic foods was calculated. 

The levels of cortisone and cortisol in aquatic foods are listed in Table 2. Cortisone was detected in all 12 aquatic food samples, with concentrations ranging from 0.72 μg/kg to 15.75 μg/kg. Cortisol was also found in all 12 aquatic food samples, with concentrations ranging from 4.90 μg/kg to 66.13 μg/kg. To illustrate the correlations between the levels of cortisone and cortisol and the geographical locations where the fish were sampled, maps of the concentration distributions of GC are shown in Figure 2. The levels of cortisone in LN, HeB, SH and FJ were below 3.0 μg/kg and were markedly lower than those in the other eight PLADs. The levels of cortisol from LN, HeB, SH and FJ were below 15.0 μg/kg and were markedly lower than those in the other seven PLADs. Among the 12 PLADs in this study, LN, HeB, SH, FJ and GX are closest to the coast. LN and HeB are adjacent to the Bohai Sea, and SH and FJ are adjacent to the East Sea. Similarly, GX is adjacent to the South Sea. In contrast, SC and NX are far from the coast. The levels of cortisone in SC and NX were 15.22 μg/kg and 15.75 μg/kg, and the levels of cortisol in SC and NX were 60.38 μg/kg and 48.56 μg/kg, respectively. The levels of cortisone and cortisol in SC and NX were higher than those in the other PLADs. However, the level of cortisone in HN was 15.22 μg/kg, and the level of cortisol in SX was 66.13 μg/kg. On the whole, the results indicated that the levels of cortisone and cortisol had obvious geographical characteristics: they were positively correlated with the distance from the coastline.

China is a vast country with many rivers and lakes and a long coastline, where people in different PLADs have different eating habits. The TDS considered the dietary habits of people in different sites, and the composite samples were prepared according to the dietary habits of each PLAD. The levels of cortisone and cortisol in aquatic foods in the TDS had obvious geographical characteristics. It was supposed that the levels of cortisone and cortisol reflected people’s eating habits of aquatic food in different PLADs. The fish consumption from the 12 provinces in this TDS are summarized in Table 3, in which the percentage of freshwater fish consumption among the total aquatic foods was calculated [16]. The highest percentage of freshwater fish consumption was 100% in SC, SX and NX, and the lowest percentages of freshwater fish consumption were 4.4% (FJ), 9.7% (HeB), 10.6% (LN) and 12.2% (SH). Similarly, the levels of cortisone and cortisol in FJ, HeB, LN and SH were low, and the levels of cortisone and cortisol in SC, SX and NX were high. Generally, the percentages of freshwater fish consumption were lower in the provinces closer to the coast than in those farther away. The correlations between the levels of cortisone and cortisol and the percentages of freshwater fish consumption were further investigated by Spearman’s correlation analysis using SPSS software. For cortisone, the correlation coefficient between the levels in aquatic food and percentages of freshwater fish consumption was 0.758 (*p* < 0.01). There was a significant positive correlation between the level of cortisone in aquatic food in the TDS and the percentage of freshwater fish consumption. There was also a significant correlation between the levels of cortisol in aquatic food in the TDS and the percentages of freshwater fish consumption (r = 0.908, *p* < 0.01). Hence, the levels of cortisone and cortisol had obvious geographical characteristics and reflected people’s eating habits.

### 3.5. Dietary Exposure to Cortisone and Cortisol in Chinese Adults through Aquatic Foods

The average consumption of aquatic foods, referred to the fourth China TDS, were used to calculate the EDI of cortisone and cortisol for an average man [16]. As shown in Figure 3, for cortisone, the lowest EDI was found in HeB (at 0.02 μg/d), while the highest intake was found in FJ (at 0.55 μg/d). The average EDI of cortisone through aquatic food was 0.16 μg/d. For cortisol, the average EDI through aquatic food was 0.72 μg/d. The lowest EDI was found in HeB (at 0.12 μg/d), while the highest intake was found in HuB (at 1.99 μg/d). The dietary intakes of cortisone and cortisol through aquatic food in the TDS were not completely consistent with the geographical distribution of GC levels. The reason for this inconsistency was the large difference in food consumption patterns between the different provinces. In particular, the consumption of aquatic foods in FJ was up to 192.92 g/d due to the rich aquatic resources there. Although the concentrations of cortisol and cortisone of the TDS sample in FJ was lower than average concentration, the EDI of cortisone via aquatic foods in FJ was the highest in the 12 provinces and the EDI of cortisol via aquatic foods in FJ was higher than the other nine provinces except for HLJ and HuB. On the other hand, the three highest concentrations of cortisone were in NX (15.75 μg/kg), SC (15.22 μg/kg) and HN (15.22 μg/kg). However, the consumptions of aquatic foods in NX (11.7 g/d), SC (11.38 g/d) and HN (5.05 g/d) were significantly lower than the average consumption (41.99 g/d). Therefore, the EDIs of cortisone vis aquatic foods in NX, SC and HN were 0.18 μg/d, 0.17 μg/d and 0.08 μg/d, respectively. The EDIs of cortisone vis aquatic foods in NX, SC and HN were close to the average and were significantly lower than in FJ. Similar to the EDI of cortisone, the EDIs of cortisol via aquatic foods in SX, SC and NX, with the highest concentration of cortisol, were lower than the average. 

Cortisol and cortisone are natural hormones in humans and animals, and they are approved for the treatment of animals without a regulation of a maximum residue limit. There are few studies on the concentration and EDI of cortisone and cortisol in the TDS. This point caught our attention due to all the of adverse effects from these hormones. In this study, however, the dietary intakes of cortisone and cortisol were calculated only for aquatic foods in the TDS. Therefore, the dietary intakes of cortisone and cortisol according to the TDS need to be focused on in the future. Further studies should examine the relationship between health and dietary intakes of cortisone and cortisol, given the multiple adverse effects of high doses or long-term intake of cortisone and cortisol.

### 3.6. Limitations

There were some limitations in this study. Firstly, the number and species of fish in the survey was somehow limited, though a significant difference was observed between the marine and freshwater fish. Larger samples sizes would increase and enrich the authenticity and reliability of the results. Secondly, the killing method was only one of the reasons why the levels of cortisone and cortisol in freshwater fish were higher than those in marine fish. The complicated source for this discrepancy should be studied in different ways.

## 4. Conclusions

In this study, the concentrations of cortisone and cortisol in fish were surveyed by random sampling and LC-MS/MS analysis. It was found that the mean levels of cortisone and cortisol in freshwater fish were significantly higher than those in marine fish. One of the possible reasons might be the different killing methods, which was proved by a Zebrafish model. 

The distribution of cortisone and cortisol levels in composite aquatic food samples from the fourth China TDS was investigated. As a result, the levels of cortisone and cortisol in aquatic food samples presented obvious geographical characteristics. Furthermore, the EDIs of cortisone and cortisol from aquatic food for Chinese adult people were also estimated, which were related to the dietary habits of different regions.

## Figures and Tables

**Figure 1 foods-11-03481-f001:**
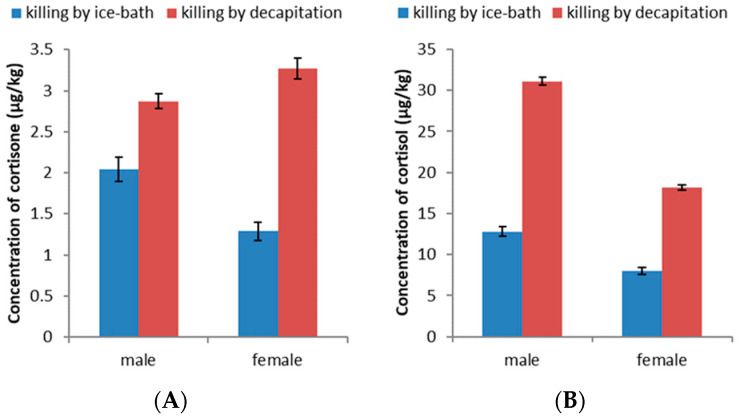
The concentrations of cortisone (**A**) and cortisol (**B**) in *Zebrafish* killed by different killing methods.

**Figure 2 foods-11-03481-f002:**
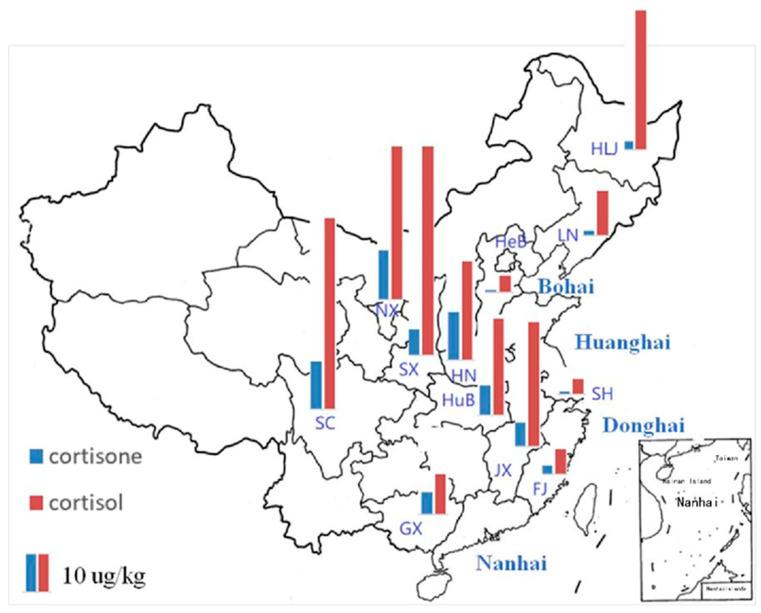
The map of concentration distribution of cortisone and cortisol in aquatic food samples from the fourth China TDS.

**Figure 3 foods-11-03481-f003:**
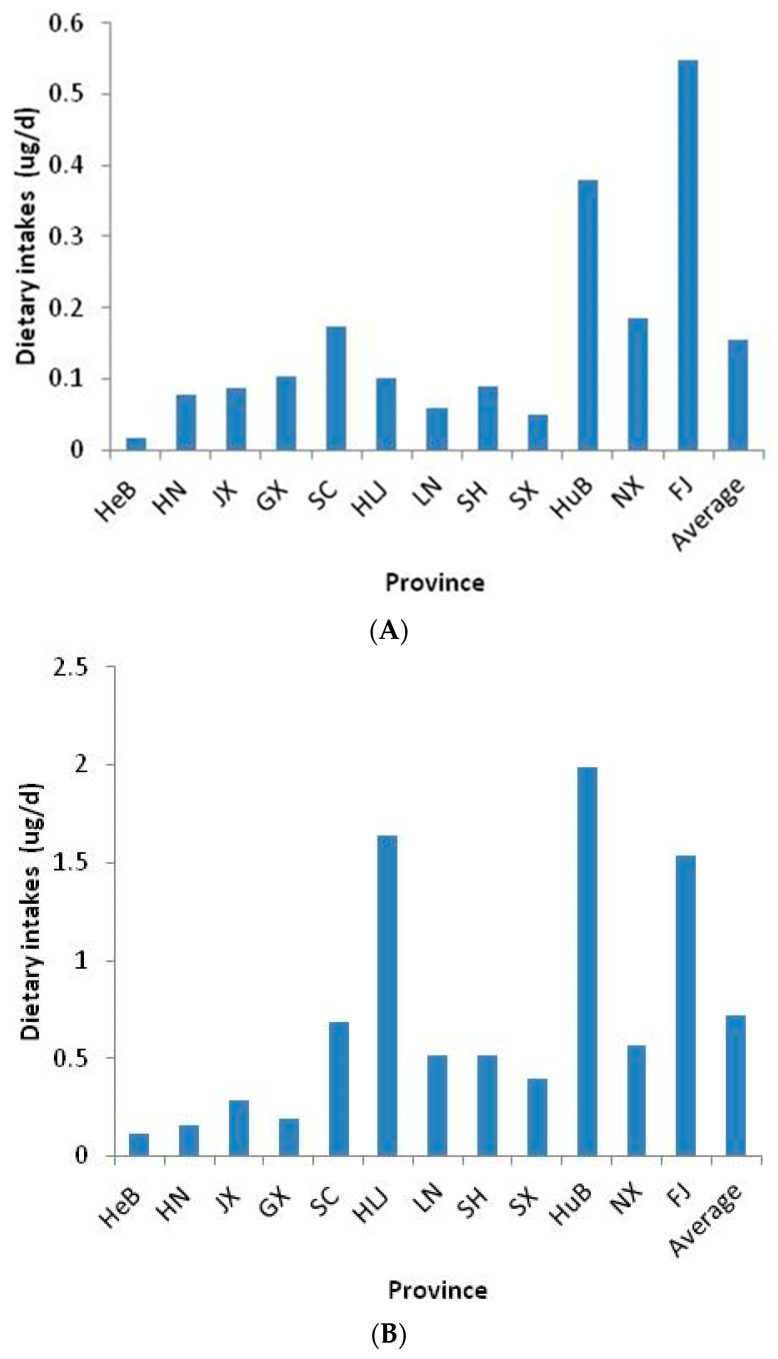
The estimated daily intakes of cortisone (**A**) and cortisol (**B**) from aquatic food from the Chinese TDS.

**Table 1 foods-11-03481-t001:** The concentrations of cortisone and cortisol in marine fish and freshwater fish.

Sample	Fish	Cortisone	Cortisol
Marine fish	Detection rate	25/26	26/26
	Maximum (μg/kg)	3.32	13.55
	Mean (μg/kg)	0.76	3.32
	Median (μg/kg)	0.53	2.19
Freshwater fish	Detection rate	29/29	29/29
	Maximum (μg/kg)	33.01	123.50
	Mean (μg/kg)	14.59	69.15
	Median (μg/kg)	11.34	62.10
Total	Mean (μg/kg)	8.05	38.03

**Table 2 foods-11-03481-t002:** Levels of cortisone and cortisol in aquatic food from the Chinese TDS (μg/kg).

Compound	Province											
	HeB	HN	JX	GX	SC	HLJ	LN	SH	SX	HuB	NX	FJ
cortisone	0.72	15.22	9.44	7.04	15.22	2.68	1.62	0.86	8.23	7.50	15.75	2.80
cortisol	5.34	31.14	30.68	12.8	60.38	44.02	14.25	4.90	66.13	39.38	48.56	7.84

**Table 3 foods-11-03481-t003:** Analysis of aquatic foods consumption from 12 PLADs in the fourth China TDS.

PLADs	Food Name	Consumption ^a^ (g)	Consumption of MF ^b^ (g)	Consumption of FF ^c^ (g)	Percent of FF (%)
HeB	*Ribbon fish*	16.97	16.97	2.12	9.7
	*Prawn*	2.74			
	*Carp*	2.12			
	*Subtotal*	21.82			
HN	*Crucian carp*	4.42	0.39	4.42	87.5
	*Ribbon fish*	0.39			
	*Small shrimp soup*	0.24			
	*Subtotal*	5.05			
JX	*Braised catfish*	1.49	0	7.66	83.1
	*Brsised carp*	5.37			
	*Braised fish fillet*	1.56			
	*Braised yellow fillet*	0.80			
	*subtotal*	9.22			
GX	*Grass carp*	8.05	4.46	10.20	69.6
	*Rice field eel*	2.15			
	*Ribbon fish*	4.46			
	*Subtotal*	14.65			
SC	*Crucian carp*	9.63	0	11.38	100
	*Grass carp*	1.75			
	*subtotal*	11.38			
HLJ	*Carp*	35.14	2.18	35.14	94.2
	*Ribbon fish*	2.18			
	*Subtotal*	37.32			
LN	*Sea crab*	8.1	15.5	3.8	10.6
	*Squid*	6.3			
	*Ribbon fish*	5.7			
	*Prawn*	5.4			
	*Carp*	3.8			
	*Scomber*	3.5			
	*Salted fish*	3.1			
	*Subtotal*	35.88			
SH	*Boiled shrimp with salt*	16.63	29.91	12.81	12.2
	*Braised flat fish*	29.91			
	*Braised shredded eel*	12.81			
	*Steamed prawn*	23.41			
	*Braised ribbon fish*	15.8			
	*Srir-fried spiral shell*	5.79			
	*Subtotal*	104.35			
SX	*Crap*	4.93	0	6.02	100
	*Grass carp*	1.10			
	*Subtotal*	6.02			
HuB	*Silver carp*	27.69	0	36.99	73.1
	*Bream fish*	9.3			
	*Lobster*	13.60			
	*Subtotal*	50.58			
NX	*Carp*	9.69	0	11.70	100
	*Crucian carp*	2.01			
	*Subtotal*	11.70			
FJ	*Razor clam*	80.03	187.28	8.64	4.4
	*Long surf clam*	13.48			
	*Ribbon fish*	15.92			
	*Little yellow croaker*	34.45			
	*cuttlefish*	13.96			
	*Sea pomfret*	14.10			
	*Sea shrimp*	15.33			
	*Grass carp*	8.64			
	*Subtotal*	195.92			

^a^ The data of consumption referenced from the fourth China TDS. ^b^ MF = marine fish. ^c^ FF = freshwater fish.

## Data Availability

The data is shown in the article.

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
