# Peer review of "Occurrence and Estimated Daily Intake of Cortisone and Cortisol in Aquatic Food from China TDS"

_foods, 2022, doi:10.3390/foods11213481_

Round 1
Reviewer 1 Report
The manuscript entitled “Occurrence and estimated daily intake of cortisone and cortisol 2 in aquatic food from China TDS” is a more important investigation for the aquatic food processing industry. The authors have surveyed and estimated the daily intake of cortisone and cortisol from aquatic food available in the seafood market. The author revealed high concentrations of cortisone and cortisol in freshwater fish compared to marine fish. The manuscript needs to improve its English, and the information presented in this paper is detailed and precise. I recommend the authors undergo a thorough minor review of the manuscript for alignment corrections.
Abstract:
Author has to rewrite “Glucocorticoids (GCs) are widely existed” to “Glucocorticoids (GCs) have widely existed”
Rewrite “The aim of the study was to survey of the occurrence” to “The study aimed to survey the occurrence”
Introduction:
Rewrite the sentence “The aim of the study presented here was to conduct a survey of the occurrence of 57 cortisone and cortisol in aquatic food and the estimated daily intake (EDI) of cortisone 58 and cortisol due to different habits of aquatic food consumption”
2. Materials and Methods
2.4.1 – Why author collected 26 marine fish? Any specific reason for this number?
Why author not collected an even number of fish samples?
The author needs to give more information on the nature of the fish, weight, size and small.
Table – 1: Author needs to provide the mean value of cortisone and cortisol concentrations.
Conclusion:
The author must provide a strong conclusion point and address other issues
Reviewer 2 Report
Although this manuscript illustrates an interesting topic it needs corrections. Unfortunately, the structure of the manuscript needs to be reorganised. The materials and methods are lacking, and the dynamics of all the experiments conducted are unclear. There is no section devoted to the protocol used for exposing zebrafish to cortisone.Despite the authors' intention to make a comparison between cortisone and cortisol levels in marine and freshwater fish, the exposition is by no means exhaustive or clear. The comparison between zebrafish treated in the laboratory(?) and the other fish used is unclear, too confusing. I understand the difference between the two but it is not explained well enough.in order to improve the quality of the manuscript ,in particular the organization of material and methods section ,i suggest the authors to take a clue from these papers(https://doi.org/10.3390/toxics10060279,https://doi.org/10.3390/toxics10040198 and cite them in the text.
The authors should correct the names of the species by writing them in italics, and standardise the text accordingly.
Lines (72-75) the authors stated : "Stock solutions (1 mg/mL) were prepared by dissolving 10 mg of an individual target compound in 10 ml methanol, and these solutions were stored at -20°C. Working solutions at serial concentrations were obtained by diluting aliquots of stock solutions with methanol". Stock solutions of which substances? it's not clear and not easy to follow all the carried procedure properly.
Lines 135: These informations are too generic.
